

# Mechanism of action of miR-15a-5p and miR-152-3p in paraquat-induced pulmonary fibrosis through Wnt/β-catenin signaling mediation

Dong Liu and Yan Guan

Weifang Medical University, Weifang, Shandong, China

## ABSTRACT

**Background:** miRNAs are small, conserved, single-stranded non-coding RNA that are typically transported by exosomes for their functional roles. The therapeutic potential of exosomal miRNAs has been explored in various diseases including breast cancer, pancreatic cancer, cholangiocarcinoma, skin diseases, Alzheimer's disease, stroke, and glioma. Pathophysiological processes such as cellular inflammation, apoptosis, necrosis, immune dysfunction, and oxidative stress are closely associated with miRNAs. Internal and external factors such as tissue ischemia, hypoxia, pathogen infection, and endotoxin exposure can trigger these reactions and are linked to miRNAs. Paraquat-induced fibrosis is a protracted process that may not manifest immediately after injury but develops during bodily recovery, providing insights into potential miRNA intervention treatments.

**Rationale:** These findings could potentially be applied for further pharmaceutical research and clinical therapy of paraquat-induced pulmonary fibrosis, and are likely to be of great interest to clinicians involved in lung fibrosis research.

**Methodology:** Through a literature review, we identified an association between miR-15a-5p and miR-152-3p and their involvement in the Wnt signaling pathway. This allowed us to deduce the molecular mechanisms underlying regulatory interactions involved in paraquat-induced lung fibrosis.

**Results:** miR-15a-5p and miR-152-3p play roles in body repair processes, and pulmonary fibrosis can be considered a form of reparative response by the body. Although the initial purpose of fibrotic repair is to restore normal body function, excessive tissue fibrosis, unlike scar formation following external skin trauma, can significantly and adversely affect the body. Modulating the Wnt/β-catenin signaling pathway is beneficial in alleviating tissue fibrosis in various diseases.

**Conclusions:** In this study, we delineate the association between miR-15a-5p and miR-152-3p and the Wnt/β-catenin signaling pathway, presenting a novel concept for addressing paraquat-induced pulmonary fibrosis.

Corresponding author
Yan Guan, guanyanwf@sina.com

## INTRODUCTION

Although paraquat (PQ) has been prohibited in numerous nations, it continues to be utilized in specific regions owing to its cost-effectiveness. The PQ-induced pulmonary fibrosis model has a higher success rate than other drug-induced models, making it conducive to pulmonary fibrosis research (*Shao et al., 2022*). MicroRNAs (miRNAs) are highly conserved single-stranded small RNAs (*Barwari, Joshi & Mayr, 2016*), operate *via* exosomal transport, and play a role in diverse signal transduction pathways (*He et al, 2018*). The Wnt/β-catenin pathway has been extensively investigated in bodily repair. The potential influence of miRNA on the Wnt/β-catenin signaling pathway in PQ-induced pulmonary fibrosis has not been explored, yet miRNA regulation of this pathway could enhance clinical treatment.

## SURVEY METHODOLOGY

We reviewed the literature surrounding miRNAs and Wnt signaling using the PubMed database (https://pubmed.ncbi.nlm.nih.gov/) to conduct an in-depth study of the molecular mechanisms underlying intersection of the two. In addition, we searched for and compared the advantages and disadvantages of traditional methods used to induce pulmonary fibrosis in animals with PQ-induced pulmonary fibrosis in the same database. Based on the above, we specified the search terms miR-15a-5p and miR-152-3p to review the relevant literature, in order to better understand the current research status involving miR-15a-5p and miR-152-3p more fully, and to discover the depth of research concerning their roles in PQ-induced pulmonary fibrosis and underlying molecular mechanisms in the Wnt/β-catenin signaling pathway.

### Clinical course and mechanism of acute PQ poisoning

PQ (N, N′-dimethyl-4,4′-bipyridine dichloride) is a herbicide. Despite being banned in many countries, it remains the most frequently utilized pesticide in developing countries because of its affordability (*Yang et al., 2022*). The lethal dose of PQ for adults is 20–40 mg/kg, and the actual dosage is 5–15 mL of an aqueous solution containing 20% PQ, which results in a high fatality rate (*Gil et al., 2014*; *Papaccio et al., 2018*).

After ingestion of PQ (20–40 mg/kg), primary aggravations typically arise due to oral ingestion or aspiration. Initially, mucosal lesions are common, involving the mouth, tongue, and pharynx. Both oesophagoscopy and gastroscopy can reveal mucosal corrosion. In severe cases, this can lead to perforation, mediastinitis, and mediastinal emphysema (*Gawarammana & Buckley, 2011*). After 1–3 days, PQ induces injury to alveolar epithelial cells, which are the primary targets in lung tissue, resulting in acute alveolitis. Patients typically experience dyspnea within 3–7 days, followed by the eventual development of secondary pulmonary fibrosis approximately 5 weeks later (*Jin & Huang, 2020*; *Shadnia et al., 2018*). Ultimately, most patients succumb to severe hypoxia caused by respiratory insufficiency (*Gawarammana & Buckley, 2011*; *Gil et al., 2014*).

Detection of PQ in the bloodstream and urine represents the gold standard for confirming PQ poisoning. Additional assessments such as high-resolution computed tomography (HRCT) of the lungs can comprehensively evaluate the severity of acute

PQ-induced lung injury. In the early stages, only a ground-glass shadow due to alveolitis may be visible, and signs of early pulmonary fibrosis may manifest after approximately 1 week (*Gawarammana & Buckley, 2011*; *Gil et al., 2014*). Additionally, arterial blood gas analysis can indicate hypoxia and lung function tests can reveal restricted lung volume (*Gawarammana & Buckley, 2011*; *Gil et al., 2014*). Serum/plasma biochemical tests, including white blood cell count, blood urea nitrogen, serum creatinine, uric acid, aspartate aminotransferase, alanine aminotransferase, and amylase levels, play a crucial role in determining prognosis (*Gawarammana & Buckley, 2011*; *Gil et al., 2014*).

Alveolar cell injury can arise from various intracellular enzymes including NADPH-cytochrome P450 reductase, xanthine oxidase, NADH-ubiquinone (REDOX) reductase, and nitric oxide synthase. PQ is enzymatically catalyzed, resulting in the generation of highly reactive oxygen radicals and alteration into an electron acceptor and donor. Consequently, this process leads to the formation of hydroxyl radicals, potent oxidants, and nitrite (ONOO) *via* the Fenton reaction. PQ also stimulates NO synthase (NOS)-mediated production of NO. The generation of highly reactive oxygen species (ROS) and nitrites may result in severe lung damage, potentially leading to respiratory failure and, ultimately, death (*Gil et al., 2014*). Second, cell signaling mediated by ROS can induce inflammation (*Chen et al., 2021*). Furthermore, lipid peroxidation, mitochondrial toxicity, NADPH oxidation, activation of nuclear factor κB (NF-κB), and apoptosis contribute to the pathophysiology of acute PQ poisoning (*Gil et al., 2014*).

The standard clinical intervention for acute PQ poisoning is gastric lavage, supplemented by hemodialysis (HD) or hemoperfusion (HP) in severe cases (*Gawarammana & Buckley, 2011*; *Gil et al., 2014*). Immunosuppressants, such as cyclophosphamide, MESNA, methylprednisolone, and dexamethasone, and antioxidants including vitamin E, vitamin C, N-acetylcysteine (NAC), desferrioxamine (DFO), and salicylic acid (SA), are frequently used in pharmacological regimens (*Gil et al., 2014*). Currently, mesenchymal stem cells derived from bone marrow, adipose tissue, and umbilical cord tissue are the current treatment options for PQ poisoning. Administration of such stem cells within 24 h of PQ poisoning in animal models has demonstrated improvements in oxidative stress responses, inflammatory responses, survival rates, and histopathological status (*Papaccio et al., 2018*).

## PQ-induced animal pulmonary fibrosis model

Currently, animal models of pulmonary fibrosis are primarily induced using drugs (*e.g.*, bleomycin or PQ) or toxins (*e.g.*, silica or asbestos). The bleomycin-induced model (*Baek et al., 2020*) is the most frequently employed; however, it has drawbacks, such as high animal mortality, complex surgical procedures, severe acute lung injury, and non-uniform lesion distribution (*Shao et al., 2022*). Asbestos and silica are rarely utilized as inducers in animal studies because of the potential human health hazards associated with mishandling (*Shao et al., 2022*).

The lungs are particularly susceptible to PQ toxicity. In a study by *Gil et al. (2014)* mice were euthanized 3 days after receiving a lethal peritoneal injection of PQ (25 mg/kg). Subsequently, the lungs, liver, and kidneys were stained with hematoxylin and eosin

(H & E) for microscopic examination. This analysis revealed pulmonary interstitial widening, inflammatory infiltration of the pulmonary parenchyma, and dense accumulation of lymphocytes in the subpleural region. Conversely, no significant histological abnormalities were observed involving the liver or kidneys (*Gil et al., 2014*). In research settings, PQ exposure can be simulated using various methods such as gastric gavage or intraperitoneal injection. These models aim to replicate the damage resulting from oral, inhalation, and dermal exposure observed in clinical PQ patients (*Shao et al., 2022*; *Stevens et al., 2021*). Animal models of PAQ-induced pulmonary fibrosis can decrease experimental animal mortality, enhance surgical success rates, and establish more suitable models for further investigation of pulmonary fibrosis (*Rychel et al., 2023*).

## Current research status of miR-15a-5p and miR-152-3p

Nucleic acids located within the nucleus are transcribed by RNA polymerase II, followed by processing involving RNA capping, splicing, and polyadenylation, resulting in the formation of a primary miRNA (pri-miRNA) (*Hill & Tran, 2021*). This primary transcript is subsequently cleaved to produce a shorter pre-miRNA (*Correia de Sousa et al., 2019*) by Drosha (an RNAse) along with its cofactor DGCR8 (*Diener, Keller & Meese, 2022*). The pre-miRNA is then transported *via* Exportin-5 through the nuclear pore, where it associates with the RISC complex containing the Argonaute protein (Ago-2) (*Ferragut Cardoso et al., 2021*; *Pritchard, Cheng & Tewari, 2012*). This association culminates in the processing of pre-miRNAs into non-coding miRNA molecules, typically 18–22 nucleotides long. These miRNAs are packaged into exosomes secreted by mesenchymal stem cells (MSCs). Upon release, they can bind to specific messenger RNA (mRNA) molecules, thereby facilitating mRNA degradation and translation. Ultimately, this process regulates the expression of other genes (*Chen et al., 2019*; *Ho, Clark & Le, 2022*; *Nasser et al., 2021*; *Yu, Odenthal & Fries, 2016*).

The mechanism of action of miR-15a-5p in atherosclerotic vessels was also investigated. Paula González-López and colleagues examined aortic and experimental atherosclerotic vessels obtained from deceased organ donors. Their findings revealed a reduction in miR-15a-5p levels within atherosclerotic samples and suggested a potential association with the NF-κB activation pathway, thus proposing its potential utility as a novel diagnostic biomarker for advanced atherosclerosis (*González-López et al., 2023*). Similarly, a study by *Li et al. (2021a)* identified the downregulation of miR-15a-5p in atherosclerotic vessels. Reduced expression of miR-15a-5p targets Chemokine (C-X3-Cmotif) ligand 1 (CX3CL1) and signal transducer and activator of transcription 3 (STAT3) signaling, thereby eliciting vascular endothelial proliferation and promoting inflammation (*Li et al., 2021a*). In another investigation concerning the viability and migratory capacity of muscle cells, miR-15a-5p was found to interact with B-cell leukemia/lymphoma 2 (Bcl-2). This study indicated that miR-15a-5p targets and downregulates Bcl-2, resulting in reduced viability and migration of muscle cells. Conversely, overexpression of Bcl-2 counteracts the effects

of miR-15a-5p, attenuating the reduction in cell viability and migration (*Peng, Wang & Li, 2022*). Furthermore, miR-15a-5p mimics also lead to increased expression of monocyte chemoattractant protein-1 (MCP-1) and matrix metallopeptidase 9 (MMP-9), an effect suppressed by the overexpression of Bcl-2 (*Peng, Wang & Li, 2022*). The impact of miRNA dysregulation on tumor development and metastasis has been extensively studied. It has been observed that miR-15a-5p exhibits decreased expression in both colon tumor tissues and associated cancer cell lines (*Li et al., 2021b*). miR-15a-5p mimics target the G1/S-specific cyclin-D1 (*CCND1*) gene to attenuate the proliferation, migration, and invasion of colon cancer cells, making it a potential molecular target for the diagnosis and treatment of this disease (*Li et al., 2021b*). The role of miR-15a-5p in conferring drug resistance in acute myeloid leukemia (AML) patients is attributed to its ability to downregulate three pro-apoptotic genes, *PDCD4, ARL2*, and *BTG2*, thereby modulating autophagy and enhancing resistance to daunorubicin-induced chemotherapy (*Bollaert et al., 2021*). However, the molecular mechanisms underlying the action of miR-15a-5p in fibrosis have not been extensively studied. In experiments involving cells *in vitro*, the upregulation of miR-15a-5p occurred by targeting vascular endothelial growth factor (VEGF), subsequently regulating epithelial-to-mesenchymal transition (EMT) in human peritoneal mesothelial cells (*He et al., 2020*). Similar results have been observed in animal experiments, revealing significant peritoneal thickening in rats undergoing peritoneal dialysis (PD), decreased miR-15a-5p expression, increased VEGF expression, and elevated collagen associated with EMT (*He et al., 2020*). In rats transfected with miR-15a-5p, there was significant thinning of the peritoneum, reduced collagen deposition, and a significant decrease in VEGF expression, which helped alleviate peritoneal fibrosis (*He et al., 2020*).

miR-152-3p is currently under-studied, but has been investigated in several diseases. Treatment of H9C2 cells with tanshinone IIA results in the upregulation of miR-152-3p and leads to anti-cardiomyocyte apoptotic effects (*Zhang et al., 2016*). miR-152-3p acts as an upstream negative regulator of phosphatase and tensin homolog (PTEN), impacting delayed healing of diabetic foot ulcer (DFU) wounds. miR-152-3p antagonists can potentially restore human umbilical vein endothelial cell (HUVEC) function and expedite wound repair, thus offering a promising target for diabetic foot ulcer (DFU) therapy (*Xu et al., 2020a*). In an *in vitro* experiment, DNA methyltransferase 1 (DNMT1) inhibited the expression of miR-152-3p by increasing methylation of the miR-152-3p promoter region, thereby suppressing mitophagy and promoting the development of heart failure (*Deng et al., 2022*). The researchers found that miR-152-3p targets $Ca^{2+}$/CaM-dependent protein kinase IIα (CaMKIIα). First, overexpression of miR-152-3p suppresses expression of CaMKIIα, subsequently affecting inflammation-related and p38 MAPK/NF-κB signaling, as well as Bcl-2/Bax/caspase-3/PARP apoptosis-related signaling in rodent models of vascular dementia (*Sun et al., 2023*). Conversely, overexpression of CaMKIIα leads to cognitive decline and neurodegenerative lesions induced by miR-152-3p (*Sun et al., 2023*). miR-152-3p has also been identified and researched in prostate cancer

(*Moya et al., 2019*), hepatocellular carcinoma (*Yin & Zhao, 2022*), thyroid carcinoma (*Rogucki et al., 2022*), glioma (*Yang et al., 2020*), and other types of malignancies.

Both miR-15a-5p and miR-152-3p interact with various signaling pathways related to inflammation, apoptosis, and fibrosis. These findings provide a basis for studying PQ-induced pulmonary fibrosis.

## Wnt signal transduction

The highly conserved *Wnt* gene, discovered in 1982, encodes a lipid-modified glycoprotein known as the Wnt protein, which consists of 350–400 amino acids (*Ng et al., 2019*). To date, 19 Wnt proteins have been identified as ligands that bind to the Frizzled (FZD) receptor, low-density lipoprotein receptor-related protein (LRP) receptor, and other receptors on the cell surface to activate signaling pathways with various functions during embryonic and adult development (*Liu, Huo & Cheng, 2023*) (Fig. 1). Binding of the Wnt protein ligand to its receptor activates the Disheveled (Dsh or Dvl) protein. The DIX, PDZ, and DEP domains of activated DSH can activate downstream pathways (Fig. 1). The signaling pathways are categorized into β-catenin-dependent (*Colozza & Koo, 2021*) and non-classical pathways based on the presence or absence of β-catenin proteins and the types of Wnt ligands (*Raslan & Yoon, 2020*).

In the canonical pathway, the destruction complex, consisting of the scaffolding protein Axin, the tumor suppressor adenomatous polyposis coli (APC), glycogen synthase kinase 3β (GSK-3β), and casein kinase 1 (CK1), regulates the cytoplasmic stability of β-catenin (Fig. 1). β-catenin binds to the destruction complex, where it is phosphorylated, ubiquitinated, and ultimately degraded by the proteasome (*Albrecht, Tejeda-Muñoz & De Robertis, 2021*; *Tian, Wang & Jang, 2022*; *Xu et al., 2020b*). In the presence of Wnt1 and its ligands such as Wnt2, Wnt3, Wnt3A, and Wnt8A, β-catenin can form complexes to prevent its degradation and precisely regulate its levels (*Lorzadeh et al., 2021*). When these Wnt ligands bind to the seven-transmembrane coiled receptor family (FZD) (*Pećina-Šlaus et al., 2023*) and the low-density lipoprotein receptor-associated protein family (LRP5/6) receptors, the destruction complex breaks down. At the same time, the DIX and PDZ domains of Dsh are activated, leading to the inhibition of Dsh protein binding to Axin and the subsequent termination of the β-catenin phosphorylation reaction within the destruction complex (*Colozza & Koo, 2021*). These two factors cause beta-catenin to accumulate in the cytoplasm. β-catenin translocates to T-cell factor/lymphoid enhancer (TCF/LEF) transcription factors in the nucleus and binds to DNA to form nuclear complexes that activate β-catenin target genes such as *Wnt*, cyclin D1 (*CCND1*), *MMP-7*, *c-Myc*, and fibronectin (*FN1*). The destruction complex is recruited to the plasma membrane to form a signaling body (*Colozza & Koo, 2021*; *Rim, Clevers & Nusse, 2022*) (Fig. 1).

Non-canonical Wnt pathways refer to the ability of Wnt5A and its ligands Wnt4, Wnt5B, Wnt6, Wnt7A, and Wnt11 to activate other signaling pathways independently of the cytoplasmic accumulation of β-catenin. This pathway is further divided into planar

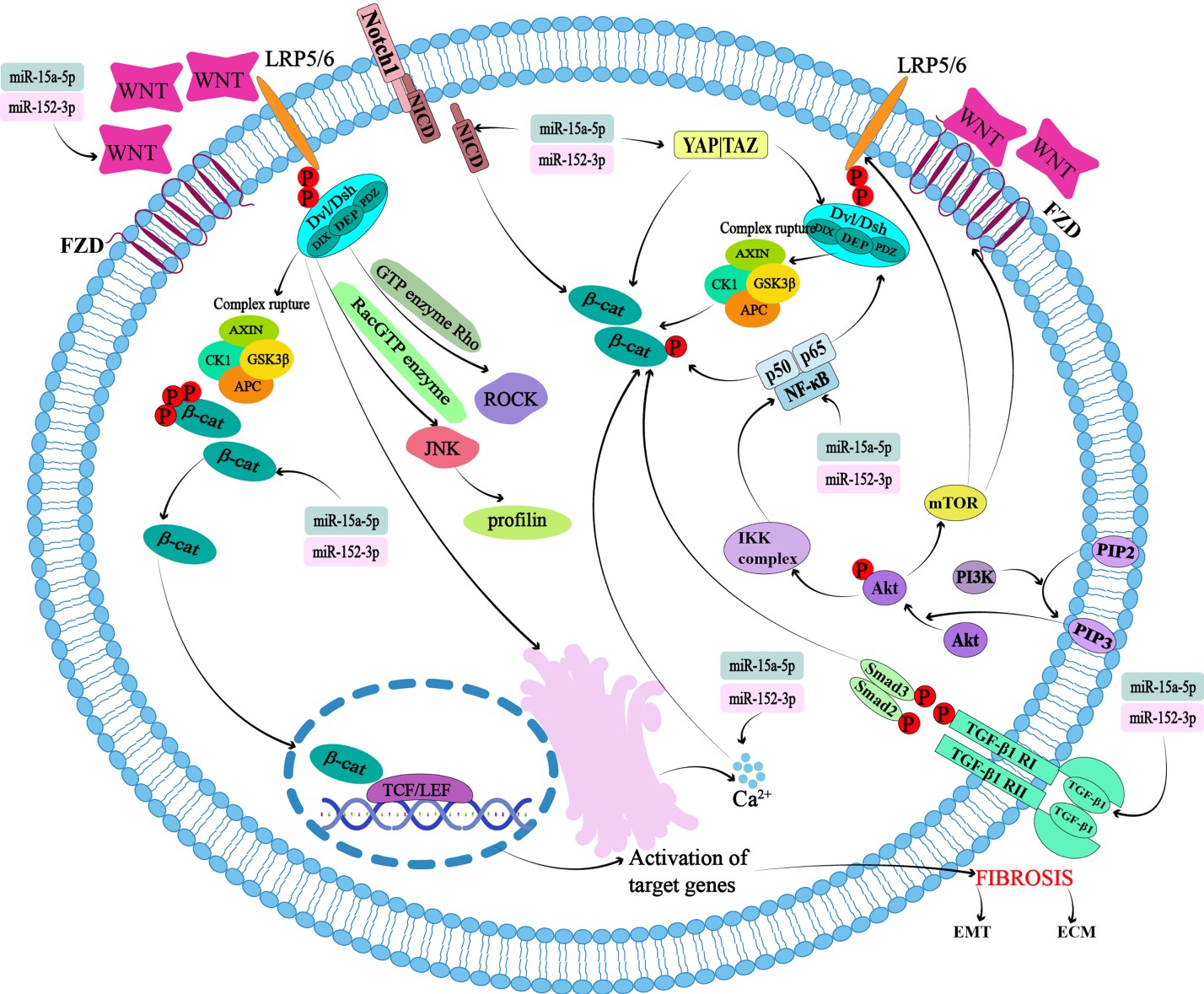

**Figure 1 Potential targets and cross-action mechanisms of miR-15a-5p and miR-152-3p in Wntβ-catenin signaling pathway in pulmonary fibrosis.** FZD, seven-transmembrane coiled receptor family; LRP, low-density lipoprotein receptor-associated protein family; Dsh or Dvl, Disheveled; APC, adenomatous polyposis coli; GSK-3β, glycogen synthase kinase 3β; CK1, casein kinase 1; TCF/LEF, T-cell factor/lymphoid enhancer transcription factors; ROCK, Rho-associated kinase; JNK, Jun kinase; ECM, extracellular matrix; EMT, epithelial-mesenchymal transition.

cell polarity (PCP) and Wnt/Ca$^{2+}$ pathways (*Raslan & Yoon, 2020*). The Wnt ligand then binds to the FZD receptor and activates Dsh. Dsh binds to the GTPase Rho, further activating Rho-associated kinase (ROCK) and resulting in cytoskeletal reorganization (Fig. 1). The DEP domain of Dsh forms a complex with Rac GTPase, which stimulates Jun kinase (JNK) and facilitates binding to actin (*Raslan & Yoon, 2020*) (Fig. 1). The Wnt/Ca$^{2+}$ pathway regulates cell activation by controlling the release of Ca$^{2+}$ from the

endoplasmic reticulum (Fig. 1). Activation of calcium/calmodulin-dependent protein kinase II cascades (CaMKII), calcineurin, or protein kinase C (PKC) regulates intracellular $Ca^{2+}$ concentrations (*Raslan & Yoon, 2020*). PKC is sensitive to $Ca^{2+}$ concentration, and thus regulates physiological and pathological changes (*Raslan & Yoon, 2020*).

## Studies on Wnt signal transduction during lung development at various sites

### Wnt signal transduction during lung development

During the embryonic phase (E9.5–E12.5), pulmonary mesodermal cell lineages are generated by pluripotent cardiopulmonary mesodermal progenitors (CPP). Wnt2 is involved, and active Wnt/β-catenin signaling is detected in gene activity assay reports (*Aros, Pantoja & Gomperts, 2021*).

During the pseudoglandular phase (E12.5–E16.5), continuous activation of β-catenin leads to distal airway dilation of surfactant protein C (Sftpc) cells, inhibiting their differentiation into secretory or ciliated cells. R-spondin2 (Rspo2) promotes embryonic lung growth and branch development by enhancing Wnt/β-catenin signaling. Wnt5a is highly expressed in alveolar endings and is essential for tracheal lengthening, cartilage ring formation, and the restriction of distal airway dilation. During branch development, Wnt2a is expressed in the distal lung submesenchymal cortex and promotes mesenchymal cell proliferation by regulating β-catenin. Wnt7b is expressed in the airway epithelium, where it plays important roles in proximal and distal pattern formation, as well as in promoting lung mesenchymal cell proliferation and pulmonary vascular growth. The deacetylase Notum can inhibit Wnt secretion and regulate the balance of tracheal development by removing Wnt lipid modifications (*Aros, Pantoja & Gomperts, 2021*).

During the luminal/vesicle phase of the embryo (E16.5-P4), the Wnt7b-BMP4 signaling axis promotes the proliferation of epithelial cells, as well as the differentiation and proliferation of stromal vascular smooth muscle cells (VSMC). The expression level of Wnt5a is highest at the distal end and promotes branching morphogenesis by inhibiting and activating Sonic Hedgehog (SHH) and Fgf10 signaling. FGF9 signals are transmitted from the epithelium and mesoderm, promoting the expression of Wnt2a in the submesodermis and stimulating stromal cell proliferation. Furthermore, downregulation of DKK1 negatively impacts the Wnt signaling pathway, resulting in premature termination of the lung epithelium. High levels of Wnt signaling promote the development of distal airway characteristics, partly through N-MYC-BMP4-FGF signaling (*Aros, Pantoja & Gomperts, 2021*).

During the alveolar formation phase (P4–P21), elevated levels of Wnt signaling prompt cell progression to alveolar Type I (ATI)-like cell lines, whereas reduced levels of Wnt signaling lead to progression to alveolar Type II (ATII)-like cell lines. The regulation of β-catenin activity is linked to abnormal hyperplasia of lung epithelial cells and proliferation of mucous cells. Furthermore, activation of Wnt signaling is linked to the proliferation of AXIN alveolar type II cells (*Aros, Pantoja & Gomperts, 2021*). These findings suggest that

the Wnt/β-catenin pathway tightly regulates the differentiation of normal respiratory epithelial cells.

### Research concerning Wnt signal transduction in lung trachea, bronchioles, and alveoli

In adult mouse models of tracheal injury, tracheal basal cells (BCs) are multifunctional stem cells capable of differentiating into ciliated, secretory, and non-ciliated columnar cells with the involvement of Wnt signaling pathways (*Raslan & Yoon, 2020*). Furthermore, the absence of GSK3 in BCs hinders phosphorylation of the N-terminal of Dsh, leading to sustained activation of Wnt signaling. The accumulated β-catenin promotes the generation of ciliated cells, thereby inhibiting the production of CLUb-like cells at the air-liquid interface. Myoepithelial cells (MECs) in the submucous glands of the trachea demonstrate a BC-like phenotype by activating Wnt/LEF1 signaling after injury and can also facilitate their replacement under normal conditions (*Raslan & Yoon, 2020*).

In a mouse model of bronchoalveolar injury, researchers have observed the expansion of bronchoalveolar stem cells (BASCs) expressing SCGB1A1 and SFTPC at the bronchoalveolar duct junction. These cells differentiate into various cell types, including ciliary, club, and ATI cells. Increased levels of Wnt ligands (Wnt3A, Wnt5A, and Wnt7B) in bronchial club cells following lung injury lead to elevated FGF10 expression in bronchial smooth muscle cells (ASMCs), thereby promoting increased cell mitosis (*Raslan & Yoon, 2020*). The knockout of RYK, a classical Wnt receptor, in club cells has been observed in naphthalene-induced lung injury, leading to an imbalance in intercellular communication due to an excessive number of mucous cells. RSPOs act as positive regulators of Wnt, and their specific receptor, G protein-coupled receptor 6 (LGR6), is expressed in ASMCs. However, genetic deletion of LGR6+ ASMCs can result in inadequate repair of damaged lungs (*Raslan & Yoon, 2020*).

Alveoli are composed of squamous cells, which are responsible for gas exchange and cover >95% of the alveolar surface, and surfactant-secreting ATII cells. Although alveolar type I (ATI) cells can regenerate and transform into alveolar type II (ATII) cells under normal conditions, neither ATI nor ATII cells can transform into one another. ATI was identified based on the expression of aquaporin 5 (AQP5), podoplanin (PDPN), RAGE, and HOPX. ATII cells express markers such as SFTPC, a lipid transporter, and the ATP-binding cassette A3 transporter.

A typical Wnt/β-catenin signal is found to be activated within ATII cells in bleomycin-induced lung injury. During the regeneration of alveoli after acute lung injury, activation of the Wnt/β-catenin signaling pathway increases AXIN2 levels in ATII cells and inhibits the differentiation of ATII cells into ATI (*Aspal & Zemans, 2020*; *Raslan & Yoon, 2020*). Thus, the differentiation of ATII and ATI cells can be regulated by controlling activity of the Wnt signaling pathway (*Raslan & Yoon, 2020*). Wnt also responds to ATII cells by secreting IL-1β, which may act as a pro-fibrotic cytokine (*Aros, Pantoja & Gomperts, 2021*). This demonstrates the significant role of Wnt signaling in cell proliferation and differentiation during lung repair to maintain the integrity of lung structure and function.

## Research on the status of the Wnt signaling pathway in pulmonary fibrosis

Beta-catenin expression is only found in endothelial and epithelial cells in normal adult lungs. Idiopathic pulmonary fibrosis (IPF) is a chronic progressive disease. Overexpression of Wnt1, Wnt5a, Wnt7b, and Wnt10b ligands, as well as phosphorylated β-cateninY489, are found in bronchial and alveolar epithelial cells and fibroblasts (*Aros, Pantoja & Gomperts, 2021*; *Chanda et al., 2019*). Wnt/β-catenin signaling has been reported in both mesenchymal and epithelial cells in patients with IPF and in murine models. Administration of Wnt signaling inhibitors has been shown to reduce fibrotic phenotypes *in vitro* and *in vivo* (*Aros, Pantoja & Gomperts, 2021*). Overexpression of Wnt5A not only stimulates the proliferation and differentiation of lung endothelial cells, but also increases the level of fibroblast growth factor-10 (FGF-10) in the mesenchyme (*Chanda et al., 2019*). The Wnt target gene, Wnt-Inducible Signaling Protein 1 (*WISP-1*), is responsible for the reduced fibrosis phenotype and improved survival in animal models. Additionally, LRP5 (Wnt co-receptor) can enhance the expression of TGF-β1, and TGF-β1 collaborates with the Wnt/β-catenin signaling pathway to enhance β-catenin-driven fibrosis. It induces epithelial-mesenchymal transition (EMT), which regulates tissue repair and deposition of extracellular matrix (ECM) components such as fibronectin and matrix metalloproteinases (MMPs). It is also associated with the downregulation of DKK-1, a negative regulator of Wnt signal transduction (*Aros, Pantoja & Gomperts, 2021*; *Chanda et al., 2019*; *Shi et al., 2017*). A clinical study comparing 10 patients with IPF and seven healthy individuals found that Wnt3A, β-catenin, and Wnt5A/B are upregulated in the lung tissue of patients with IPF (*Shi et al., 2017*). Another clinical report indicated an increase in the expression of Wnt5A and Wnt7B in human IPF lungs and that the expression of Wnt10A is linked to lower survival rates in IPF patients (*Shi et al., 2017*). Childhood bronchopulmonary dysplasia (BPD) is a rare lung disease characterized by abnormal development of the distal lung, reduced lung function, heightened inflammation, and increased fibrosis. Abnormal expression of the Wnt5a ligand and phosphorylation of β-catenin at Y489 in the lung stroma of patients with BPD (*Ota et al., 2016*). Chronic obstructive pulmonary disease (COPD) is characterized by irreversible airflow restriction caused by various factors. This is accompanied by airway inflammation, increased mucus production, peribronchiolar fibrosis, alveolar damage, and other pathological changes. The patient's lung fibroblasts secrete Wnt5a to inhibit the classical Wnt/β-catenin signaling pathway and prevent epithelial repair (*Qu et al., 2019*). Recent studies have found that aging ATII cells exhibit activation of the Wnt/beta-catenin signaling pathway, resulting in cellular senescence and pro-fibrotic changes. Abnormal expression of Wnt ligand and β-catenin Y489 protein was found in the aforementioned pulmonary diseases associated with fibrosis, indicating that Wnt signal transduction plays a crucial role in pulmonary fibrosis.

Wnt also stimulates tissue remodeling (*de Jesus Perez et al., 2014*) and cell proliferation, differentiation, and migration by regulating the expression of several genes. These include

cell cycle regulators, *Axin2*, oncogenes, Lgr5, MMPs, cyclin D1, angiogenic growth factors, and VEGF (*Villar, Zhang & Slutsky, 2019*). This is closely related to the occurrence of pulmonary fibrosis.

## Role of Wnt in pulmonary inflammation

Lung inflammation is frequently observed in patients with elevated levels of MMP and pro-inflammatory cytokines. Except for MMP-7, other members of the MMP family are not typically expressed in normal tissues. A number of experimental studies have found that activation of classical Wnt signaling can elevate MMP-2, MMP-3, MMP-7, MMP-9, and MT3-MMP protein levels in mice. This indicates that Wnt regulates MMP expression, thereby influencing lung inflammation. The upregulation of TGF-β1-induced ECM metalloproteinase inducer (EMMPRIN) in ATII cells further activates the Wnt/beta-catenin signaling pathway in fibroblasts, leading to increased production of MMP-14 (*Chanda et al., 2019*).

The Wnt/beta-catenin signaling pathway also plays a role in inflammation associated with COPD. In experiments using human bronchial epithelial cells (16HBECs) and a smoking-induced COPD mouse model, the β-catenin activator SB216763 was found to reduce the production of inflammatory factors TNF-α and IL-1β and improve inflammatory cell infiltration around the small airways affected by COPD. Levels of Tnf-α and IL-1β are reduced in bronchoalveolar lavage fluid (BALF). Conversely, using β-catenin small interfering RNA (siRNA) can enhance the production of inflammatory cytokines Tnf-α and IL-1β. This may be due to the decreased activity of peroxisome proliferator-activated receptor (PPARδ) induced by smoking and the increased phosphorylation of p38 MAPK. In contrast, β-catenin activated by SB216763 increases PPARδ activity and decreases MAPK phosphorylation. Thus, Wnt/β-catenin signaling may modulate airway inflammation in COPD through the PPARδ/p38 MAPK pathway in airway epithelial cells (*Shi et al., 2017*). The reciprocal regulation of atypical Wnt5A/B and its receptor FZD8 and TGF-β1 in fibroblasts or lung tissues of patients with COPD not only impacts lung repair but also triggers inflammation (*Shi et al., 2017*).

## Potential mechanism of Wnt signaling mediated by miR-15a-5p and miR-152-3p in PQ-induced pulmonary fibrosis

Existing studies have demonstrated that miR-15a-5p and miR-152-3p regulate fibrosis and are inextricably linked to the Wnt/β-catenin signaling pathway.

*Shen et al. (2021)* found that miR-15a-5p can regulate the EMT, and its overexpression can reduce the expression of β-catenin and Wnt//β-catenin downstream effectors in the nucleus (Fig. 1). Epithelial cells lose polarity and transform into mesenchymal cells, often referred to as EMT, which is an important component of tissue fibrosis and can occur in various organs and pathological processes involving the kidneys, liver, lungs, scarring, and heart. In a bioinformatic analysis of IPF, the Wnt signaling pathway associated with EMT suggested that miR-152-3p could target SLIT2 to promote the development of fibrosis

(*Cadena-Suárez et al., 2022*). In another study, the role of miRNA in pulmonary fibrosis was proposed through miRNA/mRNA analysis of IPF patients (*Wang et al., 2020*). It typically manifests as a process of downregulation of markers in epithelial cells (*e.g.*, E-cadherin, ZO-1, Occludin, Claudin-1, and β-catenin) and upregulation of markers in mesenchyme (*e.g.*, N-cadherin, vimentin, and fibronectin), leading to cytoskeletal remodeling and intracellular organelle dysfunction (*Luo et al., 2024*).

miR-15a-5p induces osteoblast differentiation and mineralization by inhibiting the Wnt signaling pathway inhibitor PDCD4 (programmed cell death 4) and modulating the regulators of osteoblast differentiation, alkaline phosphatase, osteocalcin, and runt-related transcription factor 2 (*Wang et al., 2021a*, *2021b*). Osteoblasts originate from MSCs and contribute to bone formation. MSCs can initiate tissue fibrosis by mobilizing a subset of specialized immunophenotypes to transform into fibroblasts (*Ghosh et al., 2023*), and also by transporting miRNAs through secretory vesicles. Wnt/β-catenin signaling has been recognized as a pathway for osteogenic differentiation (*Arya, Saranya & Selvamurugan, 2024*). Therefore, it is particularly important to study the role of MSCs carrying miRNA and its signal transduction with Wnt/β-catenin in tissue fibrosis.

Expression of miR-15a-5p is reduced in endometrial cancer cells and tissue samples from clinical patients. In *in vitro* cell experiments, miR-15a-5p overexpression can target the binding site of the Wnt3a gene 3′-UTR, thereby reducing the expression of proteins involved in cell proliferation and Wnt signaling and affecting the proliferation and stemness of HEC-1-A cells, an endometrial cancer cell line (*Wang et al., 2017*). It has been found that HOXA-AS2 releases HOXA3 through miR-15a-5p as a competitive endogenous RNA in papillary thyroid cancer and participates in the regulation of Wnt signaling-related proteins β-catenin, c-Myc and cyclin D1, regulating tumor growth and invasion (*Jiang et al., 2019*). A growing number of studies have shown that tumor proliferation and invasion are closely related to miRNAs that regulate cancer-associated fibroblasts (*Barrera et al., 2023*).

Similarly, a study by *Moya et al. (2019)* found that miR-152 is downregulated in prostate, colorectal, gastric, ovarian, breast, glioma, hepatocellular carcinoma, lung, and bladder cancer tissues compared to that in neighboring healthy tissues, which may be related to tissue progression and invasion. *In vitro* studies using different prostate cancer cell lines have shown that miR-152-3p overexpression promotes cell cycle arrest in the S and G2/M phases, which in turn impairs cell viability, cell cycle progression, and invasion potential (*Ramalho-Carvalho et al., 2018*). This may be because miR-152-3p targets the *TMEM97* and *PCNA* genes. miR-152 inhibits TCF4, a basic helix-loop-helix transcription factor, and reduces the important role of N-cadherin during EMT by increasing E-cadherin protein expression (*Jiang et al., 2020*).

Pulmonary fibrosis is also associated with EMT. The application of MSC therapy in the treatment of pulmonary fibrosis has also been studied. The roles of the miR-15a-5p, miR-152-3p, and Wnt signaling pathways in cancer are likely related to the production of cancer-associated fibroblasts. In summary, the Wnt signaling pathway mediated by miR-15a-5p and miR-152-3p is a potential therapeutic target in pulmonary fibrosis.

Secondly, the Wnt/β-catenin pathway has a cross-talk effect with other pathways. miR-15a-5p and miR-152-3p can modulate Wnt/β-catenin-mediated pulmonary fibrosis by modulating any pathway associated with Wnt/β-catenin.

It is well known that the Wnt signaling pathway promotes cell proliferation and differentiation by regulating the activity of β-catenin, and the Hippo signaling pathway inhibits cell proliferation by regulating the activity of Yap/Taz. Since 2010, researchers have gradually begun to study cross-talk between the two. Activation of the Hippo/Yap pathway in the cytoplasm restricts Dsh/Dvl nuclear translocation, thereby blocking activation of the Wnt/β-catenin pathway, leading to loss of properties of colorectal cancer stem cells and inhibition of tumor growth (*Barry et al., 2013*; *Zhao et al., 2024*) (Fig. 1). In gastrointestinal tumors such as hepatocellular carcinoma, gastric cancer, and intestinal cancer, Yap/Taz can be found to affect the Wnt/β-catenin pathway by affecting the nuclear localization of β-catenin or increasing the transcriptional activity of β-catenin (*Li, Lu & Xie, 2019*). In addition, knockout of SAV1 (SAV1 CKO), a component of the Hippo pathway in the heart, from mouse embryos revealed that β-catenin activates and enhances cardiomyocyte proliferation, thickening the ventricular wall. In contrast, β-catenin deficiency reduces SAV1 CKO-induced cardiac dilation (*Chen et al., 2020*; *Heallen et al., 2011*; *Kim & Jho, 2014*). There is currently insufficient evidence as to whether miRNA-15a-5p and miR-152-3p can affect the Hippo pathway, but noninvasive follicular thyroid neoplasms with papillary-like nuclear features (NIFTPs) have been evaluated using NanoString Technologies, and miRNA expression profiles showed that miR-152-3p is included in miRNAs with significant differences between wild-type and mutant NIFTP, as predicted by DIANA-mirPath v.3.0 of the KEGG pathway, which may be related to the Hippo signaling pathway (*Denaro et al., 2017*). Whether miR-15a-5p and miR-152-3p can modulate the Wnt/β-catenin pathway by influencing the Hippo pathway in PQ-induced pulmonary fibrosis needs to be further investigated.

Cross-talk between Wnt/β-catenin and Notch signaling was found in a study of calcific aortic valve disease (CAVD). Overexpression of the Notch1 intercellular domain (NICD) inhibits expression of β-catenin-mediated osteogenic differentiation factor (Fig. 1). Methylation of the *Notch1* promoter leads to a decrease in Notch1 expression in the nucleus of human aortic valve stromal cells (hAVICs), followed by a decrease in NICD release, thereby promoting the activation of Wnt/β-catenin signaling and expression of osteogenic differentiation factor (*Arya, Saranya & Selvamurugan, 2024*; *Zhou et al., 2019*). *Hui et al. (2019)* used an miRNA microarray approach to explore the miRNA expression profiles in juvenile idiopathic scoliosis (AIS) and non-AIS patients and found that miR-15a-5p is one of the seven most important central miRNAs. Calcium signaling pathway (*Hui et al., 2019*). Although miR-15a-5p has been shown to play a role with Notch through bioinformatic analysis, the specific point and mode of action are not known. Although the mechanism of action of miR-15a-5p and miR-152-3p in regulating the Notch pathway and affecting the components of Wnt/β-catenin in PQ-induced pulmonary fibrosis still needs to be further experimentally proved, it can be used as a potential mechanism of action to provide new ideas for the treatment of pulmonary fibrosis.
There is cross-regulation between the Wnt/β-catenin and NF-κB signaling pathways. Both β-catenin and Dvl interact with NF-κB protein. Dvl was found to interact with NF-κB p65 (*Deng et al., 2010*; *Guo et al., 2024*) (Fig. 1). β-catenin increases the stability of IκBα protein and can also bind to NF-κB subunit p50, preventing the transport of this subunit from the cytoplasm to the nucleus, thereby reducing transcriptional activity of NF-κB (*Yudhawati & Shimizu, 2023*). The investigators used bioinformatic analysis to identify that miR-15a-5p is involved in regulating the NF-κB pathway during atherosclerosis progression. Similarly, miR-15a-5p overexpression has been shown to reduce IκBα degradation and translocation of NF-κB p65 into the nucleus, thereby reducing the protein levels of IKKα, IKKβ, and p65 and NF-κB activation (*González-López et al., 2023*). This suggests that miR-15a-5p and miR-152-3p may be involved in the NF-κB and Wnt/β-catenin pathways in PQ-induced pulmonary fibrosis, and that there is signal cross-talk.

Wnt/β-catenin signaling works synergistically with TGF-β1 and contributes to pulmonary fibrosis and inflammation. TGF-β1 can also promote lung tissue inflammation, fibrotic repair, cell proliferation, and cell differentiation through its target genes, such as connective tissue growth factor (*CTGF*), α-smooth muscle actin (α-*SMA*), collagen 1a2 (*col1a2*), and PAI-2 (*serpentine-2*). This has a synergistic effect with Wnt/β-catenin (*Crosby & Waters, 2010*). TGF-β also upregulates MMP-2 or acts through the TGF-β/Smad-3 pathway (*Crosby & Waters, 2010*) (Fig. 1). TGF-β1 upregulates the Wnt/β-catenin pathway while inhibiting PPARγ, thereby affecting EMT signal transduction (*Lecarpentier et al., 2019*). circANKS1B, a circRNA derived from exons 5 to 8 of the *ANKS1B* gene hsa_circ_0007294, sponges miR-152-3p, increases the expression of transcription factor USF1, thereby up-regulating the expression of TGF-β1, activating TGF-β1/Smad signaling, and promoting EMT (*Zeng et al., 2018*). Other experiments confirmed that overexpression of miR-15a-5p leads to TGF-β3 signaling in cultured human primary retinal endothelial cells grown under high glucose conditions (*Ye, Liu & Steinle, 2017*). miR-15a-5p and miR-152-3p can affect activation of the Wnt/β-catenin signaling pathway by modulating TGF-β.

AKT can directly phosphorylate Ser552 β-catenin, and the precursor PI (4,5) P2 of PI (3,4,5) P3 can promote the nuclear translocation of β-catenin, which can then activate the Wnt/β-catenin signaling pathway. PI3K/AKT signaling activates its downstream factor, mTORC1, which in turn regulates the Wnt/β-catenin signaling pathway. In experiments using *Tsc2* knockout mice, enhanced mTORC1 activity was found to reduce cell surface FZD levels and LRP6 phosphorylation in intestinal stem cells (*Rodgers, Mitchell & Ooms, 2023*). *Yao et al. (2021)* verified that M2 macrophage-derived miR-15a can target CCND1 and block the PI3k/AKT/mTOR signaling pathway in glioma cells to inhibit tumor invasion and migration. The expression level of miR-152-3p in breast cancer tumor tissues is reduced as detected by RT-qPCR, and the PI3k/AKT signaling pathway is one of the important pathways involved in miR-152-3p targeting, based on the Reactome database (*Safi et al., 2021*). This evidence is sufficient to suggest that miR-15a-5p and miR-152-3p can bridge the cross-talk between the PI3k/AKT and Wnt/β-catenin signaling pathways.

Since the non-canonical Wnt signaling pathway is involved in calcium production, various calcium-related regulators may be involved in Wnt signaling. The concentration of

calcium in cells can affect the action of Wnt/β-catenin. Endoplasmic reticulum stress, caused by injury or ion imbalance, can activate calcium channels in the endoplasmic reticulum, leading to an increase in the concentration of calcium ions in cells. The increased calcium ions neutralize the negative charge of β-catenin and translocate it to the nucleus, initiating Wnt signaling (*Crosby & Waters, 2010*).

The mutual regulation involving certain pathways and calcium ions is also a potential target for further regulation of the Wnt/β-catenin pathway. For example, the concentration of calcium ions can also influence the activity of the PI3K/AKT pathway (*Kumari et al., 2024*; *Liang et al., 2023*), which interacts with the classical Wnt signaling pathway.

miR-15a-5p and miR-152-3p directly modulate the Wnt signaling pathway in PQ-induced pulmonary fibrosis, as well as in other cross-talk pathways.

## CONCLUSIONS

miRNAs are a group of molecules that are widely distributed in the human body and in other organisms. This article thoroughly discusses the potential roles of miR-15a-5p and miR-152-3p in regulating Wnt/β-catenin signaling in PQ-induced pulmonary fibrosis. This discussion offers a new perspective surrounding the clinical treatment of pulmonary fibrosis and presents an innovative target for the treatment of pulmonary fibrosis and other related diseases. However, due to the contextual and model-specific variability in molecular mechanisms, relevant animal experiments still needed to support this hypothesis. Therefore, the validity of this study needs to be confirmed by conducting multi-species animal experiments to fully understand the multifaceted roles of miRNAs in pulmonary fibrosis and related diseases.

In conclusion, exploring the role of miRNAs in Wnt/β-catenin signaling in PQ-induced pulmonary fibrosis presents a promising avenue for future research. This ongoing exploration will help advance our understanding of pulmonary fibrosis and related diseases and enhance diagnosis and treatment.

### Funding

This study did not receive any specific grants from funding agencies in the public, commercial, or non-profit sectors.

### Competing Interests

The authors declare that they have no competing interests.

### Author Contributions

- Dong Liu conceived and designed the experiments, performed the experiments, analyzed the data, prepared figures and/or tables, authored or reviewed drafts of the article, and approved the final draft.
- Yan Guan conceived and designed the experiments, analyzed the data, authored or reviewed drafts of the article, and approved the final draft.

## Data Availability

This is a literature review.

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
