# Peer review of "Mechanism of action of miR-15a-5p and miR-152-3p in paraquat-induced pulmonary fibrosis through Wnt/β-catenin signaling mediation"

_PeerJ, doi:10.7717/peerj.17662_

## Round 0.1 · original submission · Major Revisions

Dear authors,

Your article should go through major revision process.

Please include, as requested by Reviewer 1, the role of miR-15a-5p and miR-152-3p related to pulmonary fibrosis or Wnt signaling (most recent studies). Graphical representation of mechanisms by which miR-15a-5p and miR-152-3p could modulate the Wnt/ β-catenin signaling pathway should be included as well.

Reviewer 2 strongly advises to include a table or figure that succinctly summarizes the findings of the present study which is mandatory for the review articles.

Moreover, mechanisms through which miR-15a-5p and miR-152-3p mediate Wnt/beta-catenin pathway should be included as well parallel to graphical representation mentioned by Reviewer 1.

Also, the role of Wnt signaling in lung inflammation and fibrosis has been adequately addressed, the conclusions regarding the involvement of miRNAs extend beyond the evidence provided in the literature review (PMID: 29164582, PMID: 34672248). Please revise.

Furthermore, the connection between these miRNAs and lung fibrosis should be further elucidated and reinforced (PMID: 35743055, PMID: 31838455).

Moreover, extensive English language editing is needed. Correct reference format should be addressed.

A few other comments:
1. Line 174: the full name of STAT3 is missing or incorrectly referred.
2. Line 273: The citations for orphan references lack context.
3. Line 444-446: Please describe which factors that miR-15a-5p and miR-152-3p interact with and also regulated by Wnt signalling. There is a significant gap in establishing a connection between miR-15a-5p and miR-152-3p and Wnt signaling.

Please revise your manuscript and resubmit.
Best regards

**Language Note:** The review process has identified that the English language must be improved. PeerJ can provide language editing services - please contact us at copyediting@peerj.com for pricing (be sure to provide your manuscript number and title). Alternatively, you should make your own arrangements to improve the language quality and provide details in your response letter. – PeerJ Staff

Reviewer 1 ·

Basic reporting

The authors presented a novel hypothesis and provided strong rationale for further investigating the potential regulatory roles of miR-1a-5p and miR-152-3p in paraquat-induced pulmonary fibrosis through the Wnt/β-catenin signaling pathway. The manuscript is well-structured, with a clear introduction, methodology for literature review, and systematic presentation of relevant findings from various studies.

Authors should address some minor issues to strengthen the quality before the publication of paper:
1. From line 421, in the section that discusses the miR-15a-5p and miR-152-3p roles in various diseases, could be mor focused on the most relevant studies related to pulmonary fibrosis or Wnt signaling.

2. Authors proposed signaling mechanisms by which miR-15a-5p and miR-152-3p could modulate the Wnt/ β-catenin signaling pathway which was well supported by the literature. The authors could provide a visual representation (e.g., a schematic diagram) of the proposed mechanism, which would enhance the reader’s understanding.

Experimental design

no comment

Validity of the findings

no comment

Reviewer 2 ·

Basic reporting

The review Liu et al. aimed to investigated the MOA of miR-15a-5p and miR-152-3p in paraquat-induced pulmonary fibrosis through regulating Wnt/beta-catenin pathway. The authors described the current findings in paraquat-induced pulmonary fibrosis and miR-15a-5p/miR-152-3p. In addition, the review also introduced Wnt/beta-catenin pathway and its role during lung development, lung injury, pulmonary inflammation, and pulmonary fibrosis.

Overall, the English language should be improved to ensure that an international audience can clearly understand your text. It's recommended that the manuscript undergo a thorough proofreading by a fluent English speaker or professional editing service to avoid ambiguous language.

The introduction of the manuscript provided a brief overview of key concepts including paraquat-induced pulmonary fibrosis, miRNAs, and Wnt//beta-catenin pathway. It also briefly summarized existing research and identified gaps in the field. However, it fell short in establishing adequate relevance and significance within a broader context. Furthermore, it failed to distinctly articulate the purpose and focus of the review.

For the purpose of review, it is strongly advisable to include a table or figure that succinctly summarizes the findings of the present study.

Experimental design

The references require reformatting to adhere to the journal's recommended citation style. Instances of incorrect citation format, especially for the references with many authors such as lines 51 and 73, should be rectified accordingly. Blank spaces should be included within authors’ names.

As revealed in the title of this manuscript, the objective of this review is to investigate the MOA of miR-15a-5p and miR-152-3p in paraquat-induced pulmonary fibrosis through regulating Wnt/beta-catenin pathway. But it’s confusing that this manuscript neither dives into the the mechanisms through which miR-15a-5p and miR-152-3p mediate Wnt/beta-catenin pathway, nor elucidates their roles in paraquat-induced pulmonary fibrosis. It appears more akin to a narrative or a hypothesis than a comprehensive analysis.

Validity of the findings

In this manuscript, while the role of Wnt signaling in lung inflammation and fibrosis has been adequately addressed, the conclusions regarding the involvement of miRNAs extend beyond the evidence provided in the literature review. Additional findings concerning how miR-15a-5p and miR-152-3p mediate the Wnt/β-catenin pathway should be included. For example, Wang et al. (PMID: 29164582) demonstrated that miR-15a-5p inhibits WNT3A, while another study (PMID: 34672248) suggested that the Wnt inhibitor XAV939 can downregulate osteogenic differentiation in the miR-15a-5p/PDCD4/Wnt-dependent signaling pathway.

Furthermore, the connection between these miRNAs and lung fibrosis should be further elucidated and reinforced. For instance, a previous review (PMID: 35743055) has shed light on the role of miRNAs in pathways related to Idiopathic Pulmonary Fibrosis, and a study (PMID: 31838455) has investigated the expression of miR-152-3p in lung resident mesenchymal stem cells. It would be advantageous to provide a more comprehensive review of the context discussed to strengthen the argumentation.

A few other comments:

1. Line 174: the full name of STAT3 is missing or incorrectly referred.
2. Line 273: The citations for orphan references lack context.
3. Line 444-446: Please describe which factors that miR-15a-5p and miR-152-3p interact with and also regulated by Wnt signalling. There is a significant gap in establishing a connection between miR-15a-5p and miR-152-3p and Wnt signaling.

---

## Round 0.2 · Minor Revisions

Dear authors,

One of the reviewers requires further modifications of your manuscript. Please address the issues raised regarding Figure 1 and references. Also, please check the manuscript for several typos and grammatical errors.

**Language Note:** The Academic Editor has identified that the English language must be improved. PeerJ can provide language editing services - please contact us at copyediting@peerj.com for pricing (be sure to provide your manuscript number and title). Alternatively, you should make your own arrangements to improve the language quality and provide details in your response letter. – PeerJ Staff

Reviewer 1 ·

Basic reporting

The author answered all my questions and concerns. The article is acceptable for publication.

Experimental design

no comment

Validity of the findings

no comment

Reviewer 2 ·

Basic reporting

The authors have demonstrated significant enhancements to the manuscript's validity and depth. A few comments that need to be addressed:

1. Despite the authors' efforts to amend the citations, the absence of a space within the author's name persists, and the duplicate citation sets was included within the manuscript (Line 599-866 and Line 870-1144). It’s highly recommended to utilize reference management software such as EndNote or Mendeley to ensure accuracy and consistency in citation formatting.
2. The figures should exclusively feature validated findings discussed in the manuscript, with each arrow substantiated by referenced conclusions. Notably, in Figure 1, the association between fibrosis and MMT remains unaddressed. Furthermore, it’s recommended to revise the title of Figure 1 to enhance clarity.
3. The manuscript lacks the linkage of study results to corresponding conclusions in Line 473-477.
4. While Line 578-579 purports to offer a conclusive statement, it falls short in substantiating the roles of miR-15a-5p and miR-152-3p in other pathways in PQ-induced pulmonary fibrosis. Given the contextual and model-specific variability in molecular mechanisms, it is imperative to provide evidence demonstrating the regulatory effects of these microRNAs within a pulmonary fibrosis model, beyond their roles in other diseases or cellular models.
5. Line 174, the full name of STAT3 should be Signal transducer and activator of transcription 3.
6. Line 324, have or jave?

Experimental design

Please see above

Validity of the findings

Please see above

---

## Round 0.3 · accepted · Accept

Having addressed all the reviewer suggestions, the manuscript can now be accepted.